# OpenReview forum: "VLA-Risk: Benchmarking Vision-Language-Action Models with Physical Robustness"
_ICLR.cc/2026/Conference — Submitted to ICLR 2026_

### Official Review · Reviewer_Bp2R · 2025-10-30

**Soundness:** 3
**Presentation:** 3
**Contribution:** 3
**Rating:** 6
**Confidence:** 2

**Summary:**

This paper introduces VLA-Risk, a new benchmark for evaluating the physical robustness of Vision-Language-Action (VLA) models. Its main strength lies in its systematic construction, which defines a series of adversarial tasks across three dimensions—object, action, and space—and two modalities: vision and instruction. The experimental section validates the benchmark’s effectiveness across several mainstream VLA models and provides empirical analysis of model performance under different types of perturbations. These findings offer valuable insights for the research community and can support the future development of safer VLA systems. However, there are still some issues in the details of the manuscript.

**Strengths:**

1.Clear problem definition and well-motivated research: The paper accurately identifies the safety challenges faced by VLA models in physical deployment and clearly emphasizes the necessity for systematic robustness evaluation.

2.Systematic benchmark design: VLA-Risk is built on a principled framework. It unifies adversarial task design across three fundamental semantic dimensions (object, action, space) and two input modalities (vision and instruction), resulting in a structured and comprehensive evaluation suite.

3.Insightful Key Findings: The core finding—that "instruction-level attacks are more damaging than vision-level attacks"—is highly important and somewhat counter-intuitive. This crucial insight serves as a clear warning to the research community that focusing solely on visual robustness is insufficient, and that the language interface may present an even more critical security vulnerability.

**Weaknesses:**

1.Limited model coverage: Although the related work section introduces multiple VLA paradigms—such as Autoregressive LM-based, Diffusion/Flow-based, and Efficiency-oriented & Long-horizon VLAs—the experimental evaluation only includes three models (OpenVLA, π₀, OpenEMMA). This limited scope may restrict the generalizability of the conclusions.

2.Lack of quantitative comparison with existing attack benchmarks: While the authors provide a good overview of existing attack methods in the related work, the experiments lack direct comparisons between their proposed attacks and these existing methods on the same models and tasks.

3.Unclear operational definition of “successful execution”: For instruction-level attacks, the Attack Success Rate (ASR) is defined as the proportion where the model “incorrectly attempts to execute” an infeasible instruction. However, what constitutes an “attempt”—e.g., whether the model must generate a full action sequence or if a single commonsense-violating action suffices—requires a clearer and more operational definition, especially in long-horizon tasks.

4.Limitations of evaluation metrics in autonomous driving scenarios: In the autonomous driving domain, relying solely on Average Displacement Error (ADE) as the criterion for attack success may be insufficient. Some perturbations might not significantly alter the ADE but could still lead to critical safety violations, such as running a red light or making dangerous lane changes.

5.Potential for broader scenario coverage: The benchmark would be further strengthened by including additional scenario categories (for instance, indoor navigation, complex interaction tasks) beyond the current ones.

**Questions:**

1.Generalization Across Model Architectures: Given the diversity in VLA paradigms discussed in the related work, to what extent do you believe the identified vulnerability patterns (e.g., the higher susceptibility to instruction-level attacks) are consistent across other model families, such as diffusion-based or other autoregressive VLAs not evaluated in this study?

2.Distinction from Prior Attack Methods: While the related work effectively surveys existing attacks, could you further elaborate on the specific advantages or unique challenges that VLA-Risk's perturbation strategies address, compared to prior methods like visual patches or LLM jailbreaking techniques adapted to VLAs?

3.Definition of "Attempt" in ASR Calculation: Could you please provide more detailed criteria or decision rules for determining when an agent has "incorrectly attempted to execute" an infeasible instruction, especially for long-horizon tasks? Clarifying this operational definition would significantly enhance the reproducibility of your benchmark.

4.Safety-Oriented Metrics for Autonomous Driving: Beyond Average Displacement Error (ADE), have you considered incorporating more direct safety violation metrics (e.g., collisions, traffic rule violations) for the autonomous driving scenarios, particularly since some adversarial perturbations might induce dangerous behaviors without large trajectory deviations?

5.Benchmark Extensibility: The current benchmark covers three distinct domains. What are the primary considerations or challenges in extending VLA-Risk to include additional embodied AI scenarios, such as indoor navigation or more complex human-robot interaction tasks?

---

> ### Comment · Reviewer_5dtp · 2025-11-12
> **This review was written by AI (GPT-Zero = 100% certainty)**
>
> This review was written by AI (GPT-Zero = 100% certainty).  Not to be difficult, but it doesn't seem worth our time as reviewers to discuss a review that was so clearly not written by the reviewer. This is also disrespectful to the authors of the paper, who spent time on their submission and have the reasonable expectation of a human-written review (with the possibility of LLM editing assistance, as specified in the [reviewing guide](https://iclr.cc/Conferences/2026/ReviewerGuide)).

---

### Official Review · Reviewer_Ds39 · 2025-10-31

**Soundness:** 1
**Presentation:** 1
**Contribution:** 1
**Rating:** 0
**Confidence:** 5

**Summary:**

This paper proposes a benchmark named VLA-Risk and systematically evaluates the robustness of current VLA models from several aspects. It finally concludes that the latest VLA models are still far from being strong and robust through the empirical results.

**Strengths:**

1. The paper is easy to read and understand.

2. It provides a new benchmark, which may likely contribute to the robotics community.

**Weaknesses:**

**1. Repeated, marginal, overly simplistic idea.**

The paper basically repeats a very simple idea that has already been thoroughly explored in past attack research. The proposed attack is just an old concept directly transplanted to the VLA setting, without any substantial adaptation or new insight. Its attack scope is also narrower and less meaningful than prior attack-in-robotics work, e.g., [1].

**2. Extremely lacking in appropriate literature reviews.**

I'm very confused about why the authors totally fail to complete the literature review. One of the core attacks they rely on is the typographic attack, which has been well studied, yet they don’t cite any of the foundational or recent papers on it [2][3][4].

**3. No real-world validation.**

There are no real-robot experiments. All results are in simulation, but the simulation setup is clearly not representative of real-world conditions. Without any physical experiments or real-environment validation, their claims about robustness or safety are meaningless.


---

*References:*

[1] BadRobot: Jailbreaking Embodied LLMs in the Physical World. ICLR 2025.

[2] Unveiling Typographic Deceptions: Insights of the Typographic Vulnerability in Large Vision-Language Models. ECCV 2024.

[3] SCENETAP: Scene-Coherent Typographic Adversarial Planner against  Vision-Language Models in Real-World Environments. CVPR 2025.

[4] Manipulation facing threats: Evaluating physical vulnerabilities in end-to-end vision language action models. 2024.

**Questions:**

Please see the weaknesses.

---

### Official Review · Reviewer_1fUf · 2025-11-01

**Soundness:** 2
**Presentation:** 2
**Contribution:** 2
**Rating:** 2
**Confidence:** 3

**Summary:**

The paper introduces VLA-RISK, a benchmark to evaluate the robustness of Vision-Language-Action (VLA) models against two complementary threat classes: (i) executable visual perturbations (typographic overlays that mislabel objects, actions, or spatial cues) and (ii) infeasible instruction perturbations (commands that are grammatical but physically/semantically impossible). The benchmark spans 296 scenarios / 3,784 episodes across three domains—direct robotic manipulation, semantic reasoning, and autonomous driving—built from LIBERO, VLABench, and nuScenes, respectively. It defines episode-level attack success and, for driving, also evaluates ADE degradation. Experiments on OpenVLA, π0, and OpenEMMA show sizable robustness gaps, with instruction-level attacks averaging 63.99% ASRvs. 38.91% for vision-level attacks; OpenVLA and π0 degrade markedly on manipulation/semantic reasoning, and OpenEMMA’s trajectories worsen under both modalities. The paper argues that instruction infeasibility is especially damaging yet underexplored and provides qualitative case studies and step-overhead analyses to illustrate failure modes.

**Strengths:**

- Clear, modular threat taxonomy (object/action/space × vision/instruction) that maps neatly onto embodied pipelines and yields interpretable diagnostics across domains.
- Covers manipulation, semantic reasoning, and autonomous driving with standardized metrics and concrete scenario counts, enabling cross-model comparisons.
- Consistent performance drops (e.g., instruction ASR ~64%) across three distinct VLA families; analysis of longer execution steps, even when tasks “succeed,” highlights latent instability.

**Weaknesses:**

- ASR(inst) is defined as SR(D_pert)/SR(D_ori) “the proportion of adversarial inputs the agent incorrectly attempts to execute.” However, SR is a success metric; for infeasible instructions, “success” should arguably be refusal or safe abstention. As written, a model that safely refuses would score poorly, and a model that “succeeds” at an infeasible task raises definitional issues. The paper should redefine success/ASR for instruction infeasibility and validate with refusal/violation labels.
- Vision attacks rely on typographic overlays placed via a programmatic pipeline. While applicable, overlays like “TOUCH ME”/“SELECT LEFT” or labeling lanes/objects may not reflect common real-world nuisance patterns (lighting, occlusion, physical stickers with varied fonts/materials). There is limited analysis of transfer across fonts, sizes, transparency budgets, or physical-world tests (such as print-and-place or display attacks).
- Perturbation planning and instruction edits depend on GPT-5 prompting with constraints (e.g., edit-distance ≤ 0.1), but exact prompts, seeds, and candidate selection heuristics are only summarized; no code or release plan is specified. This raises concerns about reproducibility and potential circularity (LLM-generated adversaries versus LLM-backed victims).
- Mixing SR for manipulation/semantic tasks with ADE for driving makes headline ASR numbers hard to compare across domains. The benchmark would benefit from unified safety outcomes (e.g., violation rates) and real-robot/hardware validation or closed-loop driving to demonstrate external validity beyond open-loop ADE shifts.

**Questions:**

- How are “incorrect attempts to execute” operationalized for instruction infeasibility—do you annotate refusals/halts vs. unsafe attempts and count explicit rule violations, or is SR still task-success based? Clarifying labels and human verification would help.
- For the vision overlays, how sensitive are results to font/placement/transparency jitter and anchor localization errors (e.g., Grounding-DINO misboxes)? An ablation on perturbation budgets and stochasticity would strengthen claims about robustness gaps.
- The authors should redefine instruction robustness with safety-centric labels (refusal, safe noop, unsafe attempt, success) and compute ASR/violation rates accordingly; report inter-annotator agreement.
- Please provide attack-budget ablations (font/size/opacity/jitter), cross-font transfer, and at least a small physical test (printed stickers or tablet display) to probe sim2real.
- Harmonize evaluation across domains with common safety outcomes (e.g., collision/proximity/illegal maneuver flags) and consider closed-loop driving or hardware manipulation trials.

---

### Official Review · Reviewer_5dtp · 2025-11-05

**Soundness:** 3
**Presentation:** 2
**Contribution:** 1
**Rating:** 2
**Confidence:** 4

**Summary:**

The authors compile a testbed for assessing the safety of VLAs. The benchmark comprises manipulation and self-driving tasks. The authors show that various ways of perturbing the VLA inputs (text and images) can cause these models to operate unsafely, or to attempt to actuate physically or logically impossible actions.

**Summary of my review.** While I think this is a worthwhile problem and the compilation of tasks in this benchmark may be useful for other researchers, this paper is not yet ready for publication. Some of the claims about existing work are shaky, and the benchmark lacks depth beyond evaluating success rates (see detailed comments below). I’d recommend going back to the drawing board and expanding the scope of this benchmark, particularly to decouple alignment with poor model performance, expanding the set of tasks, and incorporating existing attacks. A good model for this would be something like HarmBench from the jailbreaking community, which incorporated existing attacks/defenses into the benchmark.

**Strengths:**

- The authors are correct that there isn’t a definitive benchmark for adversarially testing VLAs.
- The set of tasks looks like it could be useful for future research, particularly as more attention is drawn toward this topic.

**Weaknesses:**

- “Although recent studies have exposed these vulnerabilities through adversarial attacks (Wang et al., 2025; Zhou et al., 2025b; Jones et al., 2025), there is still no systematic benchmark to assess VLA’s robustness among both these two modalities rigorously” —> This may be unintentional, but this sentence sounds as if you mean to characterize these three papers as not rigorous. I’d consider rewording.
- “Prior typographic attacks have primarily targeted VLMs by exploiting scene-related textual cues embedded in images to mislead language outputs. However, this setting does not directly transfer to VLAs, whose outputs are sequences of actions rather than text.” —> I’m not sure I’m following the logic of this. Both textual and action outputs are generated autoregressively. Indeed, VLMs and VLAs often use the same LLM-based backbone (e.g., I believe OpenVLA uses a Llama backbone). It’d be worth thinking about simplifying your language. This is one of several places where simpler language would lead to clearer scientific writing.
- Re: the above point. Instead of “Instruction-Level Infeasible Perturbation category” and “Vision-Level Executable Perturbation category,” why not just call them “textual attacks” and “visual attack?” Using language to complicate what are really straightforward ideas makes the paper harder to read.
- “Compared to ours, these attacks focus more on leading to direct malicious instruction, where they ignore exploring the risk of instructions that are syntactically valid yet physically or logically impossible in the current environment” —> I’m not sure I understand the scope of this benchmark. The authors at various points seem to claim that they are interested in evaluating prompts that lead to harmful actions (“Unlike vision–language models (VLMs), where failures typically involve textual errors, mistakes in VLAs can result in dangerous physical actions.”) and at other points (like in the quote at the beginning of this bullet) they claim that they’re interested in logically/physically impossible tasks. I agree that this may pose a more implicit risk of damaging the hardware, but the authors don’t make this point clearly.
- It’d be worth tracing back the work a bit further on attacks/defenses in the context of LLMs/VLMs/VLAs. In particular, the authors claim in the intro that VLM attacks and VLA attacks are fundamentally different. The paper could be made stronger by making this point in the related work, i.e., by citing examples of attacks that apply to VLMs, and then explaining more specifically why they don’t apply to VLAs. The same is true for LLMs. I’d also recommend tracing back through papers like BadRobot and RoboPAIR, both of which tackled related problems in the context of LLM-based robotic planners, and both of which have datasets that could reasonably be adapted for a similar task to this paper. I believe the RoboPAIR data is actually derived from NuScenes as well, so there may be some overlap here for the self-driving tasks.
- “We adopt a typographic perturbation pipeline that programmatically overlays adversarial text onto scene images.” —> There’s not enough detail given here to reasonably reconstruct the pipeline used to put text onto images. Furthermore, it’s worth asking whether this is a viable threat model. It would improve the paper if the authors could write a short section describing the threat model they consider in this paper, and why it’s reasonable.
- “Second, perturbation planning selects adversarial candidates” —> It’s also unclear how the authors solve the maximization problem below this sentence. In particular, how is the constraint satisfied?
- For a benchmark paper, Section 4 is remarkably light on experiments. Is there not more that can be done here? Why not incorporate the existing attacks that are cited in the related work to generate the perturbed tasks?
- My more general feeling about this paper is: Are VLAs strong enough in the nominal case to undergo adversarial testing? That is, one thing that would improve this benchmark would be to assess whether failures are due to the model behaving unsafely (i.e., model alignment) or due to the fact that VLAs aren’t very good yet at basic instruction following. My suspicion is that both of these factors contribute. The benchmark would be much stronger if it proposed a way to decouple these two.

**Questions:**

See above.

---

### Meta-Review · Area_Chair_LyNQ · 2026-01-08

**Summary:**

This paper introduces VLA-Risk, a benchmark for evaluating the physical robustness of VLA models across different modalities and task dimensions (object, action, space). Reviewers raised significant concerns on the work's claims about existing work, the lack of survey and comparison with literature, and the lack of depth in benchmark beyond evaluating success rates.

**Reviewer Concerns:**

The reviewers raised significant concerns. The authors did not provide rebuttal.

**Reviewer Scores:**

As there is no rebuttal provided by the authors, it is unlikely the reviewers would change their scores.

---

### Decision · Program_Chairs · 2026-01-26

Reject